# Bi-Level Optimization for Pseudo-Labeling Based Semi-Supervised Learning

## Abstract

Semi-supervised learning (SSL) is a fundamental task in machine learning, empowering models to extract valuable insights from datasets with limited labeled samples and a large amount of unlabeled data. Although pseudo-labeling is a widely used approach for SSL that generates pseudo-labels for unlabeled data and leverages them as ground truth labels for training, traditional pseudo-labeling techniques often suffer from the problem of error accumulation, leading to a significant decrease in the quality of pseudo-labels and hence the overall model performance. In this paper, we propose a novel Bi-level Optimization method for Pseudo-label Learning (BOPL) to boost semi-supervised training. It treats pseudo-labels as latent variables, and optimizes the model parameters and pseudo-labels jointly within a bi-level optimization framework. By enabling direct optimization over the pseudo-labels towards maximizing the prediction model performance, the method is expected to produce high-quality pseudo-labels that are much less susceptible to error accumulation. To evaluate the effectiveness of the proposed approach, we conduct extensive experiments on multiple SSL benchmarks. The experimental results show the proposed BOPL outperforms the state-of-the-art SSL techniques.

## 1 Introduction

Despite the remarkable advances achieved by deep learning models, their widespread application has been impeded by the cost of acquiring sufficient amount of labeled data (Kingma et al., 2014; LeCun et al., 2015). In light of this challenge, semi-supervised learning (SSL) has emerged as a highly promising research area by offering the capacity to leverage a small number of labeled samples and a sufficient number of unlabeled samples for effective learning (Van Engelen & Hoos, 2020). The key for successful SSL lies in effectively exploiting the large number of unlabeled samples to remedy the shortage of the labeled data.

Many SSL methods have been developed to exploit the unlabeled data in various ways, including a variety of loss regularization based methods (Miyato et al., 2018; Zhang et al., 2020), and teacher-student model based methods (Athiwaratkun et al., 2019; Tarvainen & Valpola, 2017). One popularly adopted SSL technique is pseudo-labeling that aims to effectively expand the labeled data by generating predicted pseudo-labels for the unlabeled samples and then using them as ground-truth labels for model training. A number of recent works have explored diverse data augmentation based pseudo-labeling techniques such as MixMatch (Berthelot et al., 2019), ReMixMatch (Berthelot et al., 2020), FixMatch (Sohn et al., 2020), Dash (Xu et al., 2021) and FlexMatch (Zhang et al., 2021) to enhance the quality of the pseudo-labels and improve the model performance, while several other studies such as TSSDL (Shi et al., 2018) and LPD (Iscen et al., 2019) have explored label propagation techniques to create pseudo-labels based on the density of the local neighborhood. Although these methods achieve enhanced SSL performance, they are often prone to error accumulations from the mistakes of pseudo-labels to the afterwards model parameter updates, lacking of principled strategies to simultaneously maintain the quality of pseudo-labels and the prediction consistency between labeled and unlabeled samples. Such drawbacks can lead to poor suboptimal solutions, yielding prediction models that cannot generalize well.

In this paper, we propose a novel Bi-level Optimization method for Pseudo-label Learning (BOPL) to address the abovementioned limitations of existing pseudo-labeling techniques. Bi-level optimization provides a convenient framework for simultaneous optimization of two objectives that are at different

levels and inter-dependent. We adopt bi-level optimization to address SSL through a novel and direct pseudo-labeling design. Specifically, we treat pseudo-labels as latent variables and formulate pseudo-labeling as a bi-level optimization problem to jointly learn the pseudo-labels and model parameters at different levels through a pair of bi-level objectives. By optimizing the pseudo-labels directly, the method is expected to produce more reliable labels for the unlabeled samples. By designing the bi-level objectives properly, the quality of pseudo-labels and the prediction consistency between labeled and unlabeled data can be simultaneously enhanced to promote the generalizability of the prediction model. To validate the proposed methodology, we conduct experiments on multiple SSL benchmark datasets and compare the proposed BOPL with multiple state-of-the-art SSL methods. The experimental results demonstrate BOPL outperforms the comparison methods and achieves state-of-the-art performance. The key contribution of this paper can be summarized as follows:

- We propose a novel bi-level optimization method, BOPL, for SSL, which treats pseudo-labels as latent variables and directly optimizes them.
- We design a proper pair of bi-level objectives to ensure prediction consistency between labeled and unlabeled samples and promote generalizability of the model.
- Our empirical results validate the efficacy of the proposed BOPL by comparing with a set of state-of-the-art SSL methods.

## 2 RELATED WORKS

**Semi-Supervised Learning** With the widespread application of deep learning models, SSL has been receiving growing attentions for exploiting the unlabeled data to reduce demands for labeled samples. One popular line of this research is to develop regularization-based SSL methods that introduce additional loss terms based on the unlabeled data to enhance model training. Examples of such methods include the Π-model (Laine & Aila, 2017) and the Temporal-Ensemble (Laine & Aila, 2017), which add consistency regularizations to the loss function and leverage the exponential moving average of model predictions. Virtual Adversarial Training (VAT) (Miyato et al., 2018) is another regularization-based approach for SSL that trains a deep neural network with adversarial perturbation based regularization for all the training data samples. A more recent work, Consistency Regularization for Generative Adversarial Networks (CR-GAN) (Zhang et al., 2020), combines a generative adversarial network (GAN) with a consistency regularization term to generate pseudo-labels for the unlabeled data. Another line of SSL research centers on the teacher-student based methods, which train a student network to match the predictions of a teacher network on unlabeled data. Mean Teacher (MT) model (Tarvainen & Valpola, 2017) is a well-known method in this category. MT + Fast SWA (Athiwaratkun et al., 2019) combines Mean Teacher with Fast Stochastic Weight Averaging to further improve performance. Smooth Neighbors on Teacher Graphs (SNTG) (Luo et al., 2018) leverages a graph for the teacher model to regulate or control the distribution of features in the unlabeled samples. Interpolation Consistency Training (ICT) (Verma et al., 2022) is a regularization-based method built on teacher-student networks. It enforces consistency between the predictions on an interpolated set of unlabeled data points and the interpolation of the predictions on those points, pushing the decision boundary to low-density regions.

**Pseudo-Labeling** One early representative pseudo-labeling technique for SSL is Co-Training (Blum & Mitchell, 1998), which trains two classifiers to generate pseudo-labels on unlabeled samples for each other. Recently, more SSL studies have focused on generating good pseudo-labels to support model training. Pseudo-Label (Lee et al., 2013) generates labels for unlabeled samples based on model predictions while filtering out the low-confidence predictions. MixMatch (Berthelot et al., 2019) employs data augmentation to create multiple versions of each input data sample and obtains predictions for all of them. The predictions are then averaged to produce the pseudo-labels. Several other works, such as ReMixMatch (Berthelot et al., 2020), UDA (Xie et al., 2020), and FixMatch (Sohn et al., 2020), first generate pseudo-labels on weakly augmented samples based on confidence thresholds, and then use them as annotations for strongly augmented samples. Another set of works known as label propagation methods, such as TSSDL (Shi et al., 2018) and LPD (Iscen et al., 2019), assign pseudo-labels based on the density of the local neighborhood. DASO (Oh et al., 2022), on the other hand, blends confidence-based pseudo-labels and density-based pseudo-labels in different ways for each class. Moreover, some approaches, such as Dash (Xu et al., 2021) and FlexMatch (Zhang et al., 2021), adjust the confidence thresholds dynamically in a curriculum learning manner to

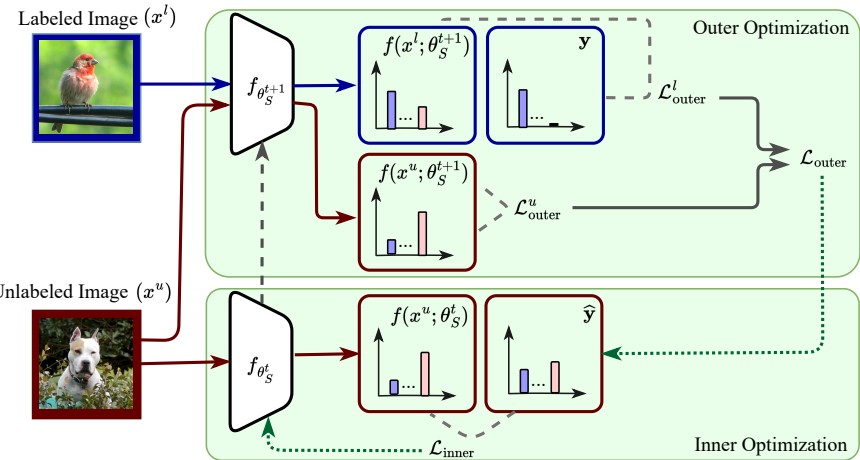

Figure 1: The proposed Bi-level Optimization framework for Pseudo-Labeling (BOPL). The inner loss $\mathcal{L}_{\text{inner}}$ is used to optimize the model parameters by leveraging the unlabeled data with pseudo-labels. The outer loss $\mathcal{L}_{\text{outer}}$ is used to optimize the pseudo-labels as latent variables by utilizing both the labeled and unlabeled samples; it treats the model parameters as a function of the pseudo-labels.

improve the quality of pseudo-labels for SSL. SimMatch (Zheng et al., 2022) uses similarity-based label propagation to improve pseudo-label quality. CoMatch (Li et al., 2021) combines consistency regularization and mutual learning for enhanced pseudo-labeling.

**Bi-level Optimization Methods**    Bi-level optimization has emerged as a powerful learning technique that allows the optimization of an inner (lower-level) objective while simultaneously optimizing an outer (higher-level) objective that depends on the solution to the lower-level problem. It has been deployed to solve various problems, including hyperparameter optimization (Pedregosa, 2016), and neural architecture search (Liu et al., 2019). Recently, some researchers have adopted bi-level optimization for SSL. In particular, Meta Pseudo-Labels (Pham et al., 2021) formulates SSL as a bi-level optimization problem that optimizes the teacher network parameters from the outer level while optimizes the student network parameters at the inner level with the pseudo-labels determined by the teacher network. Meta-Semi (Wang et al., 2020) deploys a bi-level meta optimization framework, which optimizes the weights of the unlabeled instances from the outer level based on loss on the labeled samples while learning the model parameters at the inner level by minimizing the weighted loss on unlabeled samples with predicted pseudo-labels. Although achieved state-of-the-art SSL performance, these methods still can suffer from the error accumulations due to the intermediate pseudo-label determination process. By contrast, we propose to directly optimize pseudo-labels from the outer level of a newly designed bi-level optimization framework for SSL.

## 3  PROPOSED METHOD

We consider the following SSL setting: The training dataset consists of a relatively small number of labeled samples, $\mathcal{D}^l = \{(\mathbf{x}_i^l, \mathbf{y}_i)\}_{i=1}^{N^l}$, and a large number of unlabeled samples, $\mathcal{D}^u = \{\mathbf{x}_i^u\}_{i=1}^{N^u}$, where $\mathbf{x}_i^l \in \mathcal{X}$ and $\mathbf{x}_i^u \in \mathcal{X}$ denote the $i$-th labeled and unlabeled instances respectively, and $\mathbf{y}_i$ is the corresponding one-hot label vector of $\mathbf{x}_i^l$ that indicates the class labels from the label set $\mathcal{Y}$. We assume the size of the unlabeled set greatly surpasses the labeled set: $N^u \gg N^l$. The goal is to train a classifier $f : \mathcal{X} \to \mathcal{Y}$ that generalizes well to previously unseen test data drawn from the same distribution as the training data.

In this section, we present our proposed Bi-Level Optimization method for Pseudo-label Learning (BOPL), which directly optimizes the pseudo-labels of the unlabeled samples at the outer level by treating the model parameters as a function of the pseudo-labels. The architecture of the proposed BOPL framework is illustrated in Figure 1. We present the bi-level optimization formulation in section 3.1, and provide the optimization algorithm in section 3.2. In section 4, we present a model

fine-tuning procedure given the learned pseudo-labels. Finally, we propose to further improve BOPL by incorporating interpolation consistency training into model fine-tuning in section 4.1.

## 3.1 PSEUDO-LABELING WITH BI-LEVEL OPTIMIZATION

We formulate the pseudo-labeling problem of SSL as a novel bi-level optimization problem by directly learning the pseudo-labels of unlabeled samples as latent variables through an outer level optimization, while learning the model parameters through an inner level optimization. To mitigate potential oscillations during training, we adopt a simple teacher-student concept to maintain two sets of model parameters: a student model $\boldsymbol{\theta}_S$ and a teacher model $\boldsymbol{\theta}_T$. The student model is learned directly during training, while the teacher model is updated using an exponential moving average (EMA) of the student model; at the $t$-th iteration, the update is conducted as follows:

$$\boldsymbol{\theta}_T^t = \beta \, \boldsymbol{\theta}_T^{t-1} + (1 - \beta) \, \boldsymbol{\theta}_S^t, \tag{1}$$

where $\beta \in (0, 1)$ is a momentum coefficient hyperparameter. The student and teacher models are initialized by performing pre-training on the labeled and unlabeled data for some iterations using the Mean-Teacher approach (Tarvainen & Valpola, 2017).

The bi-level loss functions of the proposed BOPL is designed as follows to induce high quality pseudo-labels while learning optimal model parameters as functions of the pseudo-labels. First, since we have a large amount of unlabeled samples, when the pseudo-labels of the unlabeled data are given, the optimal model parameters $\boldsymbol{\theta}_S^*$ can simply be learned by minimizing the standard cross-entropy loss on the pseudo-labeled data $\mathcal{D}^u$ that are originally unlabeled:

$$\mathcal{L}_{\text{inner}}(\boldsymbol{\theta}_S, \widehat{Y}) = \frac{1}{N^u} \sum\nolimits_{i=1}^{N^u} \ell_{\text{CE}}(\widehat{\mathbf{y}}_i, f(\mathbf{x}_i^u, \boldsymbol{\theta}_S)) \tag{2}$$

where $\ell_{\text{CE}}$ denotes the cross-entropy loss, $f(\mathbf{x}_i^u, \boldsymbol{\theta}_S)$ is the probabilistic prediction output of the student model for the unlabeled instance $\mathbf{x}_i^u$, and $\widehat{\mathbf{y}}_i$ denotes the pseudo-label vector for $\mathbf{x}_i^u$. As different pseudo-labels will lead to different optimal model parameters $\boldsymbol{\theta}_S^*$, we can treat $\boldsymbol{\theta}_S^*$ as a function of the pseudo-labels $\widehat{Y} = [\widehat{\mathbf{y}}_1, \cdots, \widehat{\mathbf{y}}_i, \cdots, \widehat{\mathbf{y}}_{N^u}]$. The quality of the pseudo-labels $\widehat{Y}$ determines the quality of the model parameters $\boldsymbol{\theta}_S^*$, while the latter reflects the former.

Due to the dependence of the optimal model parameters $\boldsymbol{\theta}_S^*$ on pseudo-labels, we propose to evaluate the quality of pseudo-labels by assessing the corresponding optimal model's performance. Specifically, given no separate validation set, we validate $\boldsymbol{\theta}_S^*$'s performance using the whole training set—$\mathcal{D}^l$ and $\mathcal{D}^u$. On the labeled data $\mathcal{D}^l$, we use the standard cross-entropy loss as the validation loss:

$$\mathcal{L}_{\text{outer}}^l(\boldsymbol{\theta}_S^*) = \frac{1}{N^l} \sum\nolimits_{i=1}^{N^l} \ell_{\text{CE}}(\mathbf{y}_i, f(\mathbf{x}_i^l, \boldsymbol{\theta}_S^*)). \tag{3}$$

This validation loss can naturally reflect the prediction consistency between the labeled (with true labels) and unlabeled (with pseudo-labels) samples. On the unlabeled data $\mathcal{D}^u$, we use the entropy loss as the validation loss:

$$\mathcal{L}_{\text{outer}}^u(\boldsymbol{\theta}_S^*) = \frac{1}{N^u} \sum\nolimits_{i=1}^{N^u} \ell_{\text{E}}(f(\mathbf{x}_i^u, \boldsymbol{\theta}_S^*)), \tag{4}$$

where $\ell_{\text{E}}$ denotes the entropy loss function. The entropy loss reflects the prediction uncertainty of the given model on the unlabeled samples. The overall validation loss on the training data integrates these two loss terms together:

$$\mathcal{L}_{\text{outer}}(\boldsymbol{\theta}_S^*) = \mathcal{L}_{\text{outer}}^l(\boldsymbol{\theta}_S^*) + \lambda \mathcal{L}_{\text{outer}}^u(\boldsymbol{\theta}_S^*), \tag{5}$$

where $\lambda \in [0, 1]$ is a trade-off hyperparameter. Since the labeled data have the true label information, $\lambda$ is typically set to a value smaller than 1 to give more weight to the validation loss on $\mathcal{D}^l$.

To obtain high quality pseudo-labels, we then treat the pseudo-labels $\widehat{Y}$ as latent variables and learn the pseudo-labels by minimizing the corresponding optimal model's validation loss. This leads to the following bi-level optimization problem:

$$\min_{\widehat{Y} \in \mathcal{C}} \quad \mathcal{L}_{\text{outer}}(\boldsymbol{\theta}_S^*(\widehat{Y})) \tag{6}$$

$$\text{s.t.} \quad \boldsymbol{\theta}_S^* = \arg\min_{\boldsymbol{\theta}_S} \mathcal{L}_{\text{inner}}(\boldsymbol{\theta}_S, \widehat{Y})$$

where the pseudo-label variables $\widehat{Y}$ are subject to the soft label distribution constraints, such as:

$$\mathcal{C} = \{\widehat{Y} : \ \widehat{\mathbf{y}}_i \geq 0, \ \sum_j \widehat{\mathbf{y}}_{ij} = 1, \ \forall i\}. \tag{7}$$

By optimizing the pseudo-labels directly, the proposed bi-level optimization is expected to produce more reliable high quality pseudo labels for the unlabeled samples through the outer validation loss. Meanwhile, it also provides a principled mechanism to ensure prediction consistency between the labeled and pseudo-labeled samples. These desirable properties can foreseeably enhance the generalization capacity of the subsequent prediction model.

## 3.2 Optimization Procedure

We propose to solve the bi-level optimization problem in Eq.(6) using stochastic gradient descent. The key for designing the optimization algorithm lies in deriving the gradient of the outer validation loss function $\mathcal{L}_{\text{outer}}$ regarding the pseudo-label variables $\widehat{Y} = [\widehat{\mathbf{y}}_1, \cdots, \widehat{\mathbf{y}}_i, \cdots, \widehat{\mathbf{y}}_{N^u}]$.

As the pseudo-label variables only impact the validation loss via the model parameters $\boldsymbol{\theta}_S^*$, we derive the gradient of $\mathcal{L}_{\text{outer}}$ w.r.t. each $\widehat{\mathbf{y}}_i$ using the following chain rule:

$$\nabla_{\widehat{\mathbf{y}}_i} \mathcal{L}_{\text{outer}} = \nabla_{\boldsymbol{\theta}_S^*} \mathcal{L}_{\text{outer}} \cdot \nabla_{\widehat{\mathbf{y}}_i} \boldsymbol{\theta}_S^*. \tag{8}$$

Moreover, we propose to approximate $\boldsymbol{\theta}_S^*$ by updating the model parameters $\boldsymbol{\theta}$ with a single gradient descent step over the inner loss function $\mathcal{L}_{\text{inner}}$. Specifically, at the $t$-th iteration, we set

$$\boldsymbol{\theta}_S^* = \boldsymbol{\theta}_S^{t+1} = \boldsymbol{\theta}_S^t - \alpha \nabla_{\boldsymbol{\theta}_S} \mathcal{L}_{\text{inner}}(\boldsymbol{\theta}_S^t, \widehat{Y}^t) \tag{9}$$

where $\alpha$ is the learning rate. For convenience of notation, let's define:

$$\delta(\boldsymbol{\theta}_S^{t+1}) = \nabla_{\boldsymbol{\theta}_S} \mathcal{L}_{\text{outer}}^l(\boldsymbol{\theta}_S^{t+1}) + \lambda \nabla_{\boldsymbol{\theta}_S} \mathcal{L}_{\text{outer}}^u(\boldsymbol{\theta}_S^{t+1}). \tag{10}$$

We then compute the target gradient as follows.

**Proposition 1.** *With the chain rule in Eq.(8) and the approximation in Eq.(9), the gradient of $\mathcal{L}_{outer}$ w.r.t. $\widehat{\mathbf{y}}_i$ can be expressed as:*

$$\nabla_{\widehat{\mathbf{y}}_i} \mathcal{L}_{outer} = -\alpha \cdot \delta(\boldsymbol{\theta}_S^{t+1}) \cdot \nabla_{\widehat{\mathbf{y}}_i} \nabla_{\boldsymbol{\theta}_S} \mathcal{L}_{inner}(\boldsymbol{\theta}_S^t, \widehat{Y}) \tag{11}$$

*Proof.* Given the approximation in Eq.(9), we have:

$$\nabla_{\widehat{\mathbf{y}}_i} \boldsymbol{\theta}_S^* = \nabla_{\widehat{\mathbf{y}}_i} \boldsymbol{\theta}_S^{t+1} = \nabla_{\widehat{\mathbf{y}}_i}(\boldsymbol{\theta}_S^t - \alpha \nabla_{\boldsymbol{\theta}_S} \mathcal{L}_{\text{inner}}(\boldsymbol{\theta}_S^t, \widehat{Y}^t)) = -\alpha \cdot \nabla_{\widehat{\mathbf{y}}_i} \nabla_{\boldsymbol{\theta}_S} \mathcal{L}_{\text{inner}}(\boldsymbol{\theta}_S^t, \widehat{Y}^t) \tag{12}$$

Then following the chain rule in in Eq.(8) and the definition of $\mathcal{L}_{\text{outer}}$ in Eq.(5), we can immediately derive:

$$\begin{aligned}
\nabla_{\widehat{\mathbf{y}}_i} \mathcal{L}_{\text{outer}} &= \left(\nabla_{\boldsymbol{\theta}_S} \mathcal{L}_{\text{outer}}^l(\boldsymbol{\theta}_S^{t+1}) + \lambda \nabla_{\boldsymbol{\theta}_S} \mathcal{L}_{\text{outer}}^u(\boldsymbol{\theta}_S^{t+1})\right) \cdot \nabla_{\widehat{\mathbf{y}}_i} \boldsymbol{\theta}_S^* \\
&= -\alpha \cdot \delta(\boldsymbol{\theta}_S^{t+1}) \cdot \nabla_{\widehat{\mathbf{y}}_i} \nabla_{\boldsymbol{\theta}_S} \mathcal{L}_{\text{inner}}(\boldsymbol{\theta}_S^t, \widehat{Y})
\end{aligned} \tag{13}$$

$\square$

Although it is convenient to compute the first order derivatives of the loss functions, the second order derivative $\nabla_{\widehat{\mathbf{y}}_i} \nabla_{\boldsymbol{\theta}_S} \mathcal{L}_{\text{inner}}(\boldsymbol{\theta}_S^t, \widehat{Y})$ is not easy to compute. We hence further leverage a finite difference approximation method (Bottou, 2012) for the partial derivative $\nabla_{\boldsymbol{\theta}_S} \mathcal{L}_{\text{inner}}(\boldsymbol{\theta}_S^t, \widehat{Y})$ to provide a convenient solution to the second order derivative.

**Proposition 2.** *Let $\epsilon$ be a very small constant. By using a finite difference approximation for $\nabla_{\boldsymbol{\theta}_S} \mathcal{L}_{inner}(\boldsymbol{\theta}_S^t, \widehat{Y})$, we can approximate the target gradient as follows:*

$$\nabla_{\widehat{\mathbf{y}}_i} \mathcal{L}_{outer} \approx \frac{\alpha}{2\epsilon} \Big( \log(f(\mathbf{x}_i^u; \boldsymbol{\theta}_S^+)) - \log(f(\mathbf{x}_i^u; \boldsymbol{\theta}_S^-)) \Big) \tag{14}$$

*where $\boldsymbol{\theta}_S^+ = \boldsymbol{\theta}_S^t + \epsilon \cdot \delta(\boldsymbol{\theta}_S^{t+1})$ and $\boldsymbol{\theta}_S^- = \boldsymbol{\theta}_S^t - \epsilon \cdot \delta(\boldsymbol{\theta}_S^{t+1})$.*

---

**Algorithm 1** Training Algorithm for BOPL

---

**Input**: training dataset: $\mathcal{D}^l$ and $\mathcal{D}^u$; hyperparameters; initialized model parameters: $\boldsymbol{\theta}_S, \boldsymbol{\theta}_T$;
**Output**: learned model parameters $\boldsymbol{\theta}_S$
**Set** : $\boldsymbol{\theta}_S^1 \leftarrow \boldsymbol{\theta}_S; \boldsymbol{\theta}_T^1 \leftarrow \boldsymbol{\theta}_T; \widehat{Y} = f(X^u; \boldsymbol{\theta}_T^1)$
**for** $t = 1$ to maxiters **do**
    **for** minibatch $B^u \in \mathcal{D}^u$ **do**
        Compute loss $\mathcal{L}_{\text{inner}}(\boldsymbol{\theta}_S^t, \widehat{Y})$ on $B^u$ via Eq.(2)
        Calculate $\boldsymbol{\theta}_S^{t+1}$ using Eq.(9)
        Compute $\mathcal{L}_{\text{outer}}^u(\boldsymbol{\theta}_S^{t+1})$ on $B^u$ via Eq.(4)
        Set $\mathcal{L}_{\text{outer}}^l = 0$
        **for** minibatch $B^l \in \mathcal{D}^l$ **do**
          $\mathcal{L}_{\text{outer}}^l = \mathcal{L}_{\text{outer}}^l + \frac{1}{N^l} \ell_{\text{CE}}(B^l; \boldsymbol{\theta}_S^{t+1})$
        **end for**
        Calculate $\delta(\boldsymbol{\theta}_S^{t+1})$ using Eq.(10)
        Calculate gradient $\nabla_{\widehat{\mathbf{y}}_i} \mathcal{L}_{\text{outer}}$ on $B^u$ with Eq.(14)
        Update $\widehat{Y}$ on $B^u$ with Eq.(17) and Eq.(18)
    **end for**
    $\boldsymbol{\theta}_S^{t+1} \leftarrow \boldsymbol{\theta}_S^t - \alpha \nabla_{\boldsymbol{\theta}_S} \mathcal{L}_{\text{inner}}(\boldsymbol{\theta}_S^t, \widehat{Y})$ on $\mathcal{D}^u; \quad \boldsymbol{\theta}_T^{t+1} \leftarrow \beta \boldsymbol{\theta}_T^t + (\beta - 1)\boldsymbol{\theta}_S^{t+1}$
**end for**
Fine-tune the model parameters $\boldsymbol{\theta}_S$ with the input data and the learned pseudo-labels $\widehat{Y}$

---

*Proof.* Let's use the following finite difference approximation for the partial derivative $\nabla_{\boldsymbol{\theta}_S} \mathcal{L}_{\text{inner}}(\boldsymbol{\theta}_S^t, \widehat{Y})$:

$$\nabla_{\boldsymbol{\theta}_S} \mathcal{L}_{\text{inner}}(\boldsymbol{\theta}_S^t, \widehat{Y}) \approx \frac{\mathcal{L}_{\text{inner}}(\boldsymbol{\theta}_S^+, \widehat{Y}) - \mathcal{L}_{\text{inner}}(\boldsymbol{\theta}_S^-, \widehat{Y})}{2\epsilon \cdot \delta(\boldsymbol{\theta}_S^{t+1})} \tag{15}$$

By substituting this back to Eq.(11), we can express the target gradient as:

$$\begin{aligned} \nabla_{\widehat{\mathbf{y}}_i} \mathcal{L}_{\text{outer}} &\approx -\frac{\alpha}{2\epsilon}\Big(\nabla_{\widehat{\mathbf{y}}_i} \mathcal{L}_{\text{inner}}(\boldsymbol{\theta}_S^+, \widehat{Y}) - \nabla_{\widehat{\mathbf{y}}_i} \mathcal{L}_{\text{inner}}(\boldsymbol{\theta}_S^-, \widehat{Y})\Big) \\ &= -\frac{\alpha}{2\epsilon}\Big(\nabla_{\widehat{\mathbf{y}}_i} \ell_{\text{CE}}(\widehat{\mathbf{y}}_i, f(\mathbf{x}_i^u, \boldsymbol{\theta}_S^+)) - \nabla_{\widehat{\mathbf{y}}_i} \ell_{\text{CE}}(\widehat{\mathbf{y}}_i, f(\mathbf{x}_i^u, \boldsymbol{\theta}_S^-))\Big) \\ &= \frac{\alpha}{2\epsilon}\Big(\log(f(\mathbf{x}_i^u; \boldsymbol{\theta}_S^+)) - \log(f(\mathbf{x}_i^u; \boldsymbol{\theta}_S^-))\Big) \end{aligned} \tag{16}$$

$\square$

**Update of Pseudo-Labels:** The pseudo-labels $\widehat{Y}$ can be initialized by applying prediction model with the initial teacher model parameters $\boldsymbol{\theta}_T^1$, such as $\widehat{\mathbf{y}}_i = f(\mathbf{x}_i^u; \boldsymbol{\theta}_T^1)$. Then in each iteration, we update each pseudo-label vector $\widehat{\mathbf{y}}_i$ with a gradient descent step. Moreover, to ensure a valid label distribution vector, we have to renormalize each updated pseudo-label vector to satisfy the constraints $\mathcal{C}$ in Eq.(7). In particular, we adopt the following gradient descent and renormalization process with a learning rate $\alpha$ by deploying the ReLU operator:

$$\widetilde{\mathbf{y}}_i = \frac{\text{ReLU}(\widehat{\mathbf{y}}_i - \alpha \nabla_{\widehat{\mathbf{y}}_i} \mathcal{L}_{\text{outer}})}{\sum_j \text{ReLU}(\widehat{\mathbf{y}}_i - \alpha \nabla_{\widehat{\mathbf{y}}_i} \mathcal{L}_{\text{outer}})_j} \tag{17}$$

where $\nabla_{\widehat{\mathbf{y}}_i} \mathcal{L}_{\text{outer}}$ is calculated using Eq.(14). Furthermore, to mitigate oscillations of gradient updates across iterations, we integrate the prediction outputs from the current teacher model to determine the final updated pseudo-labels. For the $t$-th iteration, we eventually update the pseudo-labels as follows:

$$\widehat{\mathbf{y}}_i = \gamma \, \widetilde{\mathbf{y}}_i + (1 - \gamma) \, f(\mathbf{x}_i^u, \boldsymbol{\theta}_T^t), \tag{18}$$

where the hyperparameter $\gamma \in [0, 1]$ determines the linear combination weight. The overall batch-wise training algorithm for the proposed BOPL is presented in Algorithm 1.

Table 1: Comparison results in terms of mean test error and standard deviation using WRN-28-2 as the backbone on CIFAR-10 and SVHN and using WRN-28-8 as the backbone on CIFAR-100.

| Dataset | CIFAR-10 | | | CIFAR-100 | | SVHN |
|---|---|---|---|---|---|---|
| Number of Labeled Samples | 250 | 1000 | 4000 | 2500 | 10000 | 1000 |
| VAT (Miyato et al., 2018) | - | $18.68_{(0.40)}$ | $11.05_{(0.31)}$ | - | - | $5.35_{(0.19)}$ |
| Mean Teacher (Tarvainen & Valpola, 2017) | $32.32_{(2.30)}$ | $17.32_{(4.00)}$ | $10.36_{(0.25)}$ | $53.91_{(0.57)}$ | $35.83_{(0.24)}$ | $5.65_{(0.45)}$ |
| ICT (Verma et al., 2022) | - | - | $7.66_{(0.17)}$ | - | - | $3.53_{(0.07)}$ |
| MixMatch (Berthelot et al., 2019) | $11.05_{(0.15)}$ | $7.75_{(0.32)}$ | $6.24_{(0.06)}$ | $39.94_{(0.37)}$ | $28.31_{(0.33)}$ | $3.27_{(0.31)}$ |
| UDA (Xie et al., 2020) | $8.82_{(1.08)}$ | $5.87_{(0.13)}$ | $4.29_{(0.07)}$ | $33.13_{(0.22)}$ | $24.50_{(0.25)}$ | $1.89_{(0.01)}$ |
| ReMixMatch (Berthelot et al., 2020) | $5.44_{(0.05)}$ | $5.73_{(0.16)}$ | $4.72_{(0.04)}$ | $27.43_{(0.31)}$ | $23.03_{(0.56)}$ | $2.83_{(0.30)}$ |
| FixMatch (Sohn et al., 2020) | $5.07_{(0.35)}$ | - | $4.26_{(0.05)}$ | $28.29_{(0.11)}$ | $22.60_{(0.12)}$ | $2.28_{(0.11)}$ |
| FlexMatch (Zhang et al., 2021) | $4.98_{(0.09)}$ | - | $4.19_{(0.01)}$ | $26.49_{(0.20)}$ | $21.90_{(0.15)}$ | $6.72_{(0.01)}$ |
| CoMatch (Li et al., 2021) | $4.91_{(0.33)}$ | - | $4.56_{(0.20)}$ | $28.37_{(0.35)}$ | $20.86_{(0.36)}$ | - |
| SimMatch (Zheng et al., 2022) | $4.84_{(0.36)}$ | - | $3.96_{(0.01)}$ | $25.07_{(0.32)}$ | $20.58_{(0.11)}$ | - |
| Meta-Semi (Wang et al., 2020) | - | $7.34_{(0.22)}$ | $6.10_{(0.10)}$ | - | - | - |
| Meta Pseudo-Labels (Pham et al., 2021) | - | - | $3.89_{(0.07)}$ | - | - | $1.99_{(0.07)}$ |
| BOPL (Ours) | $\mathbf{4.65}_{(0.27)}$ | $\mathbf{5.12}_{(0.18)}$ | $\mathbf{3.12}_{(0.08)}$ | $\mathbf{24.84}_{(0.29)}$ | $\mathbf{19.92}_{(0.27)}$ | $\mathbf{1.81}_{(0.07)}$ |
| BOPL+ICT (Ours) | $\mathbf{4.12}_{(0.26)}$ | $\mathbf{4.02}_{(0.13)}$ | $\mathbf{3.03}_{(0.08)}$ | $\mathbf{23.16}_{(0.27)}$ | $\mathbf{18.12}_{(0.24)}$ | $\mathbf{1.78}_{(0.07)}$ |

## 4 MODEL FINE-TUNING

After bi-level optimization, we can obtain high quality pseudo-labels $\widehat{Y}$ for the unlabeled samples in the training set. To ensure the model parameters are well trained given the pseudo-labels, we propose to further fine-tune the model parameters on both the labeled and pseudo-labeled data. In particular, we use the standard cross-entropy loss as the supervised loss $\mathcal{L}_{\text{sup}}$ on the labeled data $\mathcal{D}^l$ and use a mean squared error as the pseudo-label supervised loss $\mathcal{L}_{\text{pseudo}}$ on the pseudo-labeled data $(\mathcal{D}^u, \widehat{Y})$ for model fine-tuning:

$$\mathcal{L}_{\text{sup}}(\boldsymbol{\theta}_S) = \mathcal{L}_{\text{outer}}^l(\boldsymbol{\theta}_S), \qquad \mathcal{L}_{\text{pseudo}}(\boldsymbol{\theta}_S) = \frac{1}{N^u} \sum_{i=1}^{N^u} \|f(\mathbf{x}_i^u; \boldsymbol{\theta}_S) - \widehat{\mathbf{y}}_i\|_2^2 \qquad (19)$$

The mean squared error measures the stability of the model predictions on the pseudo-labeled data.

We perform model fine-tuning to maintain prediction consistency between labeled and unlabeled data by minimizing the following joint loss function:

$$\mathcal{L}_{\text{ft}} = \mathcal{L}_{\text{sup}} + \eta \mathcal{L}_{\text{pseudo}} \qquad (20)$$

where $\eta$ is a trade-off hyperparameter that balances the contribution of the two loss terms.

### 4.1 FINE-TUNING WITH ICT

Interpolation Consistency Training (ICT) (Verma et al., 2022) facilitates model training and improves model's robustness and generalizability by enforcing prediction consistency across interpolated points. We extend the idea of ICT to improve our fine-tuning procedure. As part of this approach, we generate interpolated data points by employing mix-up augmentation on unlabeled samples with their learned pseudo-labels, $(\mathcal{D}^u, \widehat{Y})$. This involves linearly combining a pair of randomly selected unlabeled samples $\mathbf{x}_i^u$ and $\mathbf{x}_j^u$ and their pseudo-labels using a mixing parameter, $\mu$, sampled from a Beta distribution:

$$\mathbf{x}^{\text{m}} = \mu \, \mathbf{x}_i^u + (1 - \mu) \, \mathbf{x}_j^u, \qquad \widehat{\mathbf{y}}^{\text{m}} = \mu \, \widehat{\mathbf{y}}_i + (1 - \mu) \, \widehat{\mathbf{y}}_j \qquad (21)$$

where $\mathbf{x}_{\text{m}}^u$ and $\widehat{\mathbf{y}}_{\text{m}}$ are the mix-up (or interpolated) sample and pseudo-label, respectively. By pairing the samples in $\mathcal{D}^u$ with a random shuffled version of it, the same number (i.e., $N^u$) of mix-up samples can be generated. We then compute the mean squared consistency loss on the mix-up samples and their corresponding pseudo-labels as follows

$$\mathcal{L}_{\text{cons}}^{\text{ICT}} = \frac{1}{N^u} \sum_{i=1}^{N^u} \|f(\mathbf{x}_i^m; \boldsymbol{\theta}_S) - \widehat{\mathbf{y}}_i^m\|_2^2 \qquad (22)$$

We fine-tune the model parameters $\boldsymbol{\theta}_S$ by replacing the $\mathcal{L}_{\text{pseudo}}$ loss in Eq.(20) with this ICT consistency loss, and refer to this extended fine-tuning approach based on BOPL as BOPL+ICT.

Table 2: Comparison results in terms of mean test error and standard deviation by using WRN-37-2 as the backbone network on STL-10.

| | Π Model | MeanTeacher | MixMatch | UDA | ReMixMatch | FixMatch | BOPL (Ours) | BOPL+ICT (Ours) |
|---|---|---|---|---|---|---|---|---|
| STL/1000 | $26.23_{(0.82)}$ | $21.43_{(2.39)}$ | $10.41_{(0.61)}$ | $7.66_{(0.56)}$ | $5.23_{(0.45)}$ | $5.17_{(0.63)}$ | $\mathbf{4.93}_{(0.54)}$ | $\mathbf{4.11}_{(0.52)}$ |

## 5 EXPERIMENTS

### 5.1 EXPERIMENTAL SETUP

**Datasets:** We conducted comprehensive experiments on three commonly used image classification benchmarks: CIFAR-10, CIFAR-100 (Krizhevsky et al., 2009), SVHN (Netzer et al., 2011) and STL-10 (Coates et al., 2011). Following previous works, on each dataset we randomly select a subset of samples with equal sizes from each class as labeled data and keep the remaining samples unlabeled. We conducted experiments on CIFAR-10 with $\{250, 1,000, 2,000, 4,000\}$ labeled samples, on CIFAR-100 with 2,500, 4,000, and 10,000 labeled samples, on SVHN with 1000 and 500 labeled samples, and on STL-10 with 1,000 images as the labeled data.

**Implementation Details**: We maintained a fair comparison by adopting the same backbones, training parameters, and initial input preprocessing as prior works. Detailed implementation information is available in Appendix A.

### 5.2 COMPARISON RESULTS

We compare the proposed BOPL approach with a great set of state-of-the-art SSL methods, including Π-model (Laine & Aila, 2017), Mean Teacher (Tarvainen & Valpola, 2017), VAT MixMatch (Berthelot et al., 2019), FixMatch (Sohn et al., 2020), ReMixMatch (Berthelot et al., 2020), FlexMatch (Zhang et al., 2021), UDA (Xie et al., 2020), CoMatch (Li et al., 2021), SimMatch (Zheng et al., 2022), Meta Pseudo-Labels (Pham et al., 2021), ICT (Verma et al., 2022). We use CNN-13, WRN-28-2, WRN-28-8 and WRN-37 separately as the backbone networks to conduct comprehensive experiments and provide fair comparisons with these previous SSL methods.

Table 1 reports the comparison results on all the three datasets by using WRN-28-2 as the backbone network on CIFAR-10 and SVHN and using WRN-28-8 as the backbone network on CIFAR-100. BOPL outperforms all the other state-of-the-art comparison methods such as Mean Teacher, VAT, MixMatch, Meta-Semi, SimMatch, CoMatch, ReMixMatch and Meta Pseudo-Labels in terms of mean test error across all cases. In particular, when using only 250 and 1000 labeled samples on CIFAR-10, BOPL achieves impressive low test error of 4.56% and 5.12%, respectively. Furthermore, BOPL achieves mean test errors of 24.84% and 19.92% on CIFAR-100 with 2500 and 10000 labeled samples, respectively. Lastly, on SVHN, BOPL attains a mean test error of 1.81% using 1000 labeled samples. These outcomes demonstrate the strong performance of BOPL across varying labeled sample sizes and datasets.

Table 2 provides a comprehensive comparison of various SSL methods on the STL-10 dataset, utilizing WRN-37 as the backbone network. With a fixed number of 1000 labeled samples, our proposed method, BOPL, achieves remarkable results with a mean test error of 4.93%. These results outperform previous state-of-the-art methods, including FixMatch, ReMixMatch, and UDA, showcasing again the effectiveness of our proposed approach.

We have included additional experiments employing a CNN-13 backbone on the CIFAR-10, CIFAR-100, and SVHN datasets in Appendix B for a more detailed comparative analysis.

Overall, the proposed BOPL approach outperforms the state-of-the-art SSL methods across different experimental settings adopted in many previous works, demonstrating its effectiveness in semi-supervised image classification tasks with limited labeled samples. The consistent superiority of BOPL across multiple benchmark datasets underscores its potential as an effective tool for SSL.

### 5.3 ABLATION STUDY

We conducted an ablation study to investigate the contribution of different components of BOPL on CIFAR-100 by using CNN-13 as the backbone network. In particular, we compared the full model

Table 3: Ablation study results in terms of mean test error and standard deviation on CIFAR-100 by using CNN-13 as the backbone.

| | BOPL | $-$w/o EMA | $-$w/o $\mathcal{L}_{\text{outer}}^u$ | $-$w/o $\mathcal{L}_{\text{outer}}^l$ | $-$w/o fine-tuning | $-$w/o pre-training |
|---|---|---|---|---|---|---|
| CIFAR-100/4000 | $\mathbf{36.78}_{(0.29)}$ | $40.01_{(0.54)}$ | $38.76_{(0.49)}$ | $45.34_{(0.63)}$ | $42.12_{(0.55)}$ | $44.15_{(0.57)}$ |
| CIFAR-100/10000 | $\mathbf{28.92}_{(0.18)}$ | $31.75_{(0.43)}$ | $30.12_{(0.42)}$ | $39.92_{(0.52)}$ | $35.80_{(0.45)}$ | $37.52_{(0.43)}$ |

Table 4: Impact of pseudo-label update methods. Results are in terms of mean test error and standard deviation on CIFAR-100 by using CNN-13 as the backbone.

| | ReLU+Teacher (BOPL) | ReLU | Softmax | Softmax+Teacher |
|---|---|---|---|---|
| CIFAR-100/4000 | $\mathbf{36.78}_{(0.29)}$ | $37.11_{(0.38)}$ | $38.12_{(0.41)}$ | $37.57_{(0.30)}$ |
| CIFAR-100/10000 | $\mathbf{28.92}_{(0.18)}$ | $30.25_{(0.27)}$ | $31.95_{(0.39)}$ | $29.43_{(0.28)}$ |

BOPL with the following variants: (1) " $-$w/o EMA", which drops the teacher model parameters by disabling the EMA update; (2) " $-$w/o $\mathcal{L}_{\text{outer}}^u$", which drops the unlabeled data from the outer loss; (3) " $-$w/o $\mathcal{L}_{\text{outer}}^l$", which drops the labeled data from the outer loss; and (4) " $-$w/o fine-tuning", which drops fine-tuning. The comparison results are reported in Table 3. We can see the variant " $-$w/o $\mathcal{L}_{\text{outer}}^l$" produces the largest test error increase, which indicates the labeled data is critical for assessing the quality of the pseudo-labels and ensuring prediction consistency between labeled and pseudo-labeled data under the proposed bi-level optimization framework. Meanwhile, all the variants have higher test errors than the full model BOPL, which suggests all the components contribute to the effective performance of BOPL.

## 5.4 IMPACT OF PSEUDO-LABEL UPDATE

Our proposed bi-level optimization method aims to learn high quality pseudo-labels to support SSL. It is important to properly update the pseudo-labels to ensure valid pseudo-label vectors along the stochastic gradient descent training process of the BOPL framework. We conduct experiments on CIFAR-100 with both 4000 and 10000 labeled samples by using CNN-13 as the backbone network to investigate the impact of pseudo-label update strategies.

In particular, we considered the following pseudo-label update strategies: (1) "ReLU+Teacher", which is the pseudo-label update strategy we adopted for BOPL. It first applies ReLU based normalization on the updated pseudo-label vector via Eq.(17) and then integrates the teacher model's prediction via Eq.(18). (2) "ReLU", which only uses the ReLU based update in Eq.(17) and drops the teacher model predictions. (3) "Softmax", which uses the softmax normalization in the following form: $\widehat{\mathbf{y}}_i = \text{softmax}(\tau(\widehat{\mathbf{y}}_i - \alpha \nabla_{\widehat{\mathbf{y}}_i} \mathcal{L}_{\text{outer}}))$, where $\tau$ is the sharpening hyper-parameter and is set to 7 in the experiments. (4) "Softmax+Teacher", which further integrates the teacher model's prediction after applying softmax normalization, similar to Eq.(18). The comparison results are reported in Table 4. We can see integrating teacher model's prediction can help reduce test errors in both cases—"ReLU+Teacher" and "Softmax-Teacher", which highlights the effectiveness of combining the learned pseudo-labels with the teacher model predictions. Meanwhile, "ReLU" normalization works better than "Softmax" normalization, and our proposed pseudo-label update strategy "ReLU+Teacher" produces the best performance.

## 6 CONCLUSION

In this paper, we proposed BOPL, a novel bi-level optimization approach for Semi-Supervised Learning, which produces high-quality pseudo-labels for unlabeled data by directly learning pseudo-labels in the outer level of the bi-level optimization, while jointly optimizing model parameters at the inner level. This approach provides a structured and principled framework for overcoming the error accumulation problem of other pseudo-labeling techniques ensuring a more accurate and reliable learning process. We conducted comprehensive experiments using diverse datasets, including CIFAR-10, CIFAR-100, SVHN, and STL-10, with varying numbers of labeled samples and different backbone networks. The proposed BOPL method produces consistent and remarkable performance gains over a great set of previous SSL methods. These results validate the efficacy of the proposed approach in diverse scenarios and underline its potential for practical applications.

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

Table 5: Comparison of mean test error and standard deviation by using CNN-13 on CIFAR-10 and CIFAR-100.

| Dataset | CIFAR-10 | | | CIFAR-100 | |
|---|---|---|---|---|---|
| Number of Labeled Samples | 1000 | 2000 | 4000 | 4000 | 10000 |
| Supervised | $39.95_{(0.75)}$ | $27.67_{(0.12)}$ | $20.42_{(0.21)}$ | $58.31_{(0.89)}$ | $44.56_{(0.30)}$ |
| Supervised + MixUp (Zhang et al., 2018) | $31.83_{(0.65)}$ | $24.22_{(0.15)}$ | $17.37_{(0.35)}$ | $54.87_{(0.07)}$ | $40.97_{(0.47)}$ |
| Π-model (Laine & Aila, 2017) | $28.74_{(0.48)}$ | $17.57_{(0.44)}$ | $12.36_{(0.17)}$ | $55.39_{(0.55)}$ | $38.06_{(0.37)}$ |
| Temp-ensemble (Laine & Aila, 2017) | $25.15_{(1.46)}$ | $15.78_{(0.44)}$ | $11.90_{(0.25)}$ | - | $38.65_{(0.51)}$ |
| Mean Teacher(Tarvainen & Valpola, 2017) | $21.55_{(0.53)}$ | $15.73_{(0.31)}$ | $12.31_{(0.28)}$ | $45.36_{(0.49)}$ | $35.96_{(0.77)}$ |
| VAT (Miyato et al., 2018) | $18.12_{(0.82)}$ | $13.93_{(0.33)}$ | $11.10_{(0.24)}$ | - | - |
| SNTG (Luo et al., 2018) | $18.41_{(0.52)}$ | $13.64_{(0.32)}$ | $10.93_{(0.14)}$ | - | $37.97_{(0.29)}$ |
| Learning to Reweight (Ren et al., 2018) | $11.74_{(0.12)}$ | - | $9.44_{(0.17)}$ | $46.62_{(0.29)}$ | $37.31_{(0.47)}$ |
| MT + Fast SWA (Athiwaratkun et al., 2019) | 15.58 | 11.02 | 9.05 | - | $33.62_{(0.54)}$ |
| ICT (Verma et al., 2022) | $12.44_{(0.57)}$ | $8.69_{(0.15)}$ | $7.18_{(0.24)}$ | $40.07_{(0.38)}$ | $32.24_{(0.16)}$ |
| Meta-Semi (Wang et al., 2020) | $10.27_{(0.66)}$ | $8.42_{(0.30)}$ | $7.05_{(0.27)}$ | $37.61_{(0.56)}$ | $30.51_{(0.32)}$ |
| Meta-Semi + ICT (Wang et al., 2020) | $9.29_{(0.62)}$ | $7.05_{(0.12)}$ | $6.42_{(0.18)}$ | $37.12_{(0.59)}$ | $29.68_{(0.05)}$ |
| BOPL (Ours) | $\mathbf{8.74}_{(0.32)}$ | $\mathbf{6.90}_{(0.17)}$ | $\mathbf{5.98}_{(0.16)}$ | $\mathbf{36.78}_{(0.31)}$ | $\mathbf{28.92}_{(0.18)}$ |
| BOPL + ICT (Ours) | $\mathbf{8.54}_{(0.31)}$ | $\mathbf{6.72}_{(0.16)}$ | $\mathbf{5.79}_{(0.14)}$ | $\mathbf{36.44}_{(0.29)}$ | $\mathbf{28.80}_{(0.14)}$ |

Table 6: Comparison of mean test error and standard deviation by using CNN-13 on SVHN.

| Dataset | SVHN | |
|---|---|---|
| Number of Labeled Samples | 500 | 1000 |
| VAT (Miyato et al., 2018) | - | $5.42_{(0.00)}$ |
| Π-model (Laine & Aila, 2017) | $6.65_{(0.53)}$ | $4.82_{(0.17)}$ |
| Temp-ensemble (Laine & Aila, 2017) | $5.12_{(0.13)}$ | $4.42_{(0.16)}$ |
| Mean Teacher (Tarvainen & Valpola, 2017) | $4.18_{(0.27)}$ | $3.95_{(0.19)}$ |
| ICT (Verma et al., 2022) | $4.23_{(0.15)}$ | $3.89_{(0.04)}$ |
| SNTG (Luo et al., 2018) | $3.99_{(0.24)}$ | $3.86_{(0.27)}$ |
| Meta-Semi (Wang et al., 2020) | $4.12_{(0.21)}$ | $3.92_{(0.11)}$ |
| Meta-Semi + ICT (Wang et al., 2020) | $3.98_{(0.09)}$ | $3.77_{(0.05)}$ |
| BOPL (Ours) | $\mathbf{3.43}_{(0.06)}$ | $\mathbf{3.26}_{(0.05)}$ |
| BOPL + ICT (Ours) | $\mathbf{3.05}_{(0.05)}$ | $\mathbf{2.90}_{(0.04)}$ |

## A  IMPLEMENTATION DETAILS

Following previous works (Luo et al., 2018; Tarvainen & Valpola, 2017), we adopted random $2 \times 2$ translation and random horizontal flip to augment the training set. To maintain fair comparisons with a great number of previous studies on the multiple datasets, we conducted comprehensive experiments by using **four types** of backbone networks: a 13-layer CNN (CNN-13), a Wide-RestNet-28-2 (WRN-28-2) (Zagoruyko & Komodakis, 2016), a Wide-RestNet-37-2 (WRN-37), and a Wide-RestNet-28-8 (WRN-28-8). The WRN models are chosen based on previous works for comparability (Berthelot et al., 2019). For training CNN-13, we employed the SGD optimizer with a Nesterov momentum (Nesterov, 1983) of 0.9, an L2 regularization coefficient of 1e-4 for CIFAR-10 and CIFAR-100 datasets and 5e-5 for SVHN, and an initial learning rate $\alpha$ of 0.1. To schedule the learning rate, we utilized the cosine learning rate annealing technique (Loshchilov & Hutter, 2017; Verma et al., 2022), which reduces the learning rate in a cosine-like fashion to help the model converge at better minima. The WRN-28-2 model was trained using SGD as the optimizer as well. An L2 regularization coefficient of 5e-4 and an initial learning rate of 0.01 were employed. For the WRN-37-2 model, the training configuration includes the SGD optimizer, an L2 regularization coefficient of 5e-4, and an initial learning rate of 0.01. Lastly, for the WRN-28-8 model, the training setup involves the SGD optimizer, an L2 regularization coefficient of 1e-3, and an initial learning rate of 0.01. Specifically for BOPL, we set the batch size to 128, $\lambda = $ 1e-2, $\epsilon = $ 1e-2, $\gamma = 0.5$, $\beta = 0.999$, and $\eta = 1$. We pre-train the model for 50 epochs using the Mean-Teacher algorithm and then proceed to train BOPL for 400 epochs. Finally, we fine-tune the model for 100 epochs using both the labeled data and the unlabeled data with learned pseudo-labels. For each experiment, we repeat five independent runs and report the mean test error with standard deviation.

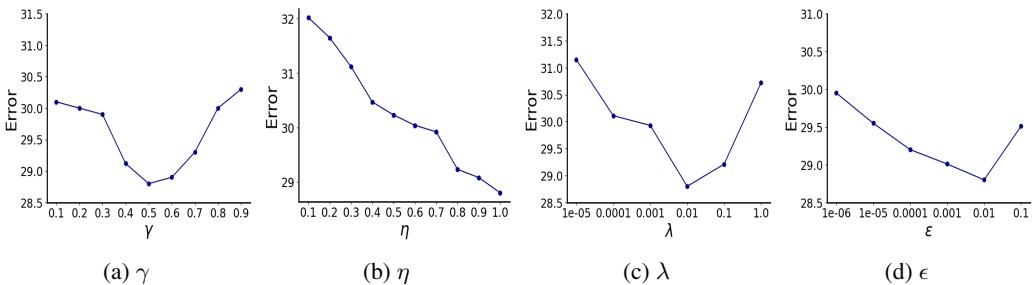

Figure 2: Sensitivity analysis for four hyper-parameters $\gamma$, $\eta$, $\lambda$ and $\epsilon$ on CIFAR-100 using 10000 labeled samples.

## B ADDITIONAL COMPARISON RESULTS

We have performed additional experiments using a CNN-13 backbone comparing our approach to Supervised + MixUp (Zhang et al., 2018), $\Pi$-model (Laine & Aila, 2017), Temp-ensemble (Laine & Aila, 2017), Mean Teacher (Tarvainen & Valpola, 2017), VAT (Miyato et al., 2018), SNTG (Luo et al., 2018), Learning to Reweight (Ren et al., 2018), MT + Fast SWA (Pham et al., 2021), ICT (Verma et al., 2022), and Meta-Semi and Meta-Semi + ICT (Wang et al., 2020).

Table 5 reports the comparison results on CIFAR-10 with 4000, 2000, and 1000 labeled samples and on CIFAR-100 with 10000 and 4000 labeled samples when CNN-13 is used as the backbone network. We can see the proposed BOPL approach outperforms all the other SSL methods on both CIFAR-10 and CIFAR-100 for all cases with different numbers of labeled samples. Compared to the state-of-the-art method, Meta-Semi, BOPL reduces the test error by 1.53%, 1.52%, and 1.07% on CIFAR-10 with 1000, 2000, and 4000 labeled samples, respectively. On CIFAR-100 with 10000 and 4000 labeled samples, BOPL reduces the test error from Meta-Semi by 0.83% and 1.59% respectively. In addition, BOPL + ICT further boosts the performance, while both BOPL an BOPL+ICT outperform Meta-Semi+ICT, which benefits from the same ICT improvement technique.

Table 6 reports the comparison results on the SVHN dataset with CNN-13 as the backbone network. BOPL achieves the lowest test error of 3.43% and 3.26% for 500 and 1000 labeled samples respectively, outperforming all the other comparison SSL methods. In particular, BOPL outperforms the second-best performing method (without the ICT extension), Meta-Semi, by 0.66% and 0.69% for 500 and 1000 labeled samples, respectively. In line with our findings on CIFAR-10 and CIFAR-100, the extended BOPL+ICT exhibits superior performance over Meta-Semi+ICT.

## C HYPER-PARAMETER SENSITIVITY ANALYSIS

We conduct sensitivity analysis for the proposed BOPL method over four hyperparameters: $\gamma$—the trade-off hyperparameter for pseudo-label update, $\eta$—the trade-off hyperparameter for the fine-tuning loss terms, $\lambda$—the trade-off hyperparameter in the outer loss of bi-level optimization, and $\epsilon$—the hyperparameter for gradient approximation in Proposition 2. We conduct experiments on CIFAR-100 using 10000 labeled instances by testing a range of different values for each of the four hyper-parameters independently.

As shown in Figure 2, the results indicate that the optimal value for $\gamma$ is 0.5. Specifically, small values of $\gamma$ will diminish the influence of the optimization learned pseudo-labels, while large values will reduce the impact of teacher predictions, resulting in higher test errors. It is therefore essential to balance the impact of both the learned pseudo-labels and teacher model output by selecting an appropriate value in the middle for $\gamma$. Our experiments also reveal that decreasing the value of $\eta$ leads to larger test errors, which further validates the effectiveness of the learned pseudo-labels for unlabeled samples. Additionally, we observe that larger values of $\lambda$ can lead to less dominant $\mathcal{L}_{\text{outer}}^{l}$ and higher test errors. Conversely, very low values of $\lambda$ result in a negligible contribution of $\mathcal{L}_{\text{outer}}^{u}$ and increased high test errors. It is important to determine a suitable small value for $\lambda$—e.g., around 0.01—to achieve good performance. Finally, it is important to set $\epsilon$ to a suitable small value close to zero to provide proper gradient approximation, as high values like 0.1 can cause significant errors and very small values can cause numerical gradient issues and high test errors as well.

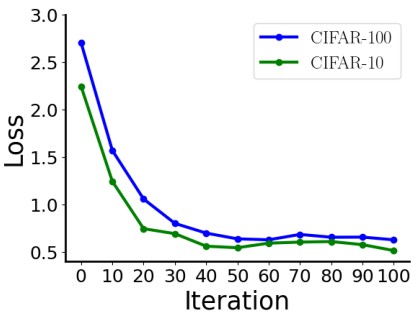

Figure 3: The outer loss values v.s. the training iterations on CIFAR-100 and CIFAR-10.

## D    EMPIRICAL CONVERGENCE ANALYSIS

Figure 3 presents the curves of the outer loss values, i.e., the validation loss on the training data, with the increasing of the training iterations on CIFAR-100 and CIFAR-10. The CIFAR-100 dataset shows a higher initial loss, indicating the more challenging nature of this dataset. We can see that as training progresses, the outer loss for the bi-level optimization decreases on both datasets. After about 50 iterations, the loss values become very flat, demonstrating empirical convergence.

