# OpenReview forum: "Bi-Level Optimization for Pseudo-Labeling Based Semi-Supervised Learning"
_ICLR.cc/2024/Conference — Submitted to ICLR 2024_

### Official Review · Reviewer_akKq · 2023-10-17

**Soundness:** 3 good
**Presentation:** 3 good
**Contribution:** 3 good
**Rating:** 6
**Confidence:** 3

**Summary:**

This paper designs a bi-level optimization method for semi-supervised learning (SSL). In the outer loop, it optimizes the pseudo-labels by minimizing the validation loss. In the inner loop, it optimizes the model parameter by empirical risk minimization. Extensive experiments validate the effectiveness of the proposed approach.

**Strengths:**

- The paper is well written and easy to understand.
- The proposed method is simple yet effective, and the introduction of bi-level optimization is novel in the SSL literature.
- To solve the optimization problem in the outer loop, the approximation approach is sound and effective. The entire algorithmic process is simple and interesting.

**Weaknesses:**

- The convergence of the proposed method is not revealed either theoretically or empirically. I think both theoretical and empirical analysis can be done to analyze the convergence problem. Similar theoretical analysis of convergence can be found in many bi-level optimization papers, and introducing it can improve the paper. Besides, it is also important to present the convergence figure of parameters, i.e. pseudo-labels in this paper. For example, the empirical study of the convergence property can be referred to Shu et al. (2019).

- Since the approach explicitly introduces the validation data sets in the training phase, it is not clear whether the comparison with the previous method is still fair. Previous methods at most used the validation set for tuning hyperparameters. Authors should explicitly discuss this issue. They should also discuss the number of validation data for the method, because I am afraid that the imparity problem gets worse with more validation data.

I am willing to raise my score if my concerns can be addressed.

***
Reference:
- Shu et al., Meta-Weight-Net: Learning an Explicit Mapping For Sample Weighting, NeurIPS 2019.

**Questions:**

- Can the method converge quickly?
- Is the comparison with previous SSL methods still fair?
- What's the impact of the number of validation data?

---

> ### Author Response · Authors · 2023-11-20
>
> We sincerely appreciate the time and effort the reviewer has dedicated to reviewing our work.
> * **About convergence analysis**
>
> Thanks for the suggestion. We added empirical convergence analysis to Appendix D of the revised paper.
>
> * **About validation data and fair comparison**
>
> We **did not** introduce separate validation datasets. The term of “validation loss” in the paper refers to the outer loss, which is used to assess/validate the quality of pseudo-labels through the model parameters. It is computed on the whole training set—the labeled training data $\mathcal{D}^l$ and the unlabeled training data $\mathcal{D}^u$ for SSL (please see Eq.(3), Eq.(4) and Eq.(5)), which are available for all the SSL methods. We did not introduce validation datasets.  We used **the same data splits and inputs** as previous studies, ensuring fair comparisons.

---

> > ### Comment · Reviewer_akKq · 2023-11-20
> > **Follow-up discussions**
> >
> > Thanks for the additional experiments. I am still concerned about the second problem. As far as I know, it seems that most bi-level optimization problems in machine learning (such as meta-learning) use different datasets for inner and outer optimization. I am not sure whether using the same dataset is problematic. Could you provide references using the similar strategies?

---

> ### Author Response · Authors · 2023-11-20
>
> Thanks for the quick response and follow up discussion.
>
> Bi-level optimization and meta-learning represent two distinct problems. Bi-level optimization involves nested optimization problems, where the solution to a lower-level problem serves as a constraint for an upper-level problem.  In bi-level optimization, the same data is used for two levels of loss in a different way. For example, in our BOPL the inner objective is cross-entropy loss on unlabeled data with pseudo-labels, and the outer objective is the cross-entropy loss on labeled data plus entropy loss on the unlabeled data. In contrast, meta-learning, also known as "learning to learn," focuses on designing models that improve their learning process from accumulated experience. In meta-learning the validation set needs to be different. As references to papers that use bi-level optimization with the same strategy as ours, you can see the works that we discussed in the related works of our paper including [1][2][3][4].
>
> [1] Fabian Pedregosa. Hyperparameter optimization with approximate gradient. In International
> conference on machine learning (ICML), 2016.
>
> [2] Hanxiao Liu, Karen Simonyan, and Yiming Yang. DARTS: Differentiable architecture search. In International Conference on Learning Representations (ICLR), 2019
>
> [3] Hieu Pham, Zihang Dai, Qizhe Xie, and Quoc V Le. Meta pseudo labels. In IEEE/CVF Conference on Computer Vision and Pattern Recognition (CVPR), 2021
>
> [4] Yulin Wang, Jiayi Guo, Shiji Song, and Gao Huang. Meta-semi: A meta-learning approach for
> semi-supervised learning. arXiv preprint arXiv:2007.02394, 2020

---

> ### Comment · Reviewer_akKq · 2023-11-20
> **Further discussion**
>
> Thanks for the reply. I am still not convinced.
>
> Sorry for my unclear presentation. I mean BLO-based meta-learning methods, which are the mainstream meta-learning methods. I think this kind of methods is a special case of BLO.
>
> Besides, I found that [3,4] are both using different datasets for the outer and inner optimization problems. They use labeled and unlabeled training datasets separately. They are not the same as yours.

---

> > ### Author Response · Authors · 2023-11-20
> >
> > Sorry for the confusion. We answered the previous question regarding using the same data in the context of whether an additional validation set is introduced. From this perspective, same as our proposed approach, both Meta pseudo-label[3] and Meta-Semi[4] use data from **the same SSL training set (labeled data $\mathcal{D}^l$ and unlabeled data $\mathcal{D}^u$)** for the bi-level objectives, without additional validation set.
> >
> > The **specific data** used for the bi-level objectives of course **are different**. In our BOPL approach, we used the labeled data and the unlabeled data (without labels) for the outer objective (i.e., outer loss), and used the unlabeled data with pseudo-labels for the inner objective. Meta pseudo-label [3] uses the unlabeled data for the inner objective and the labeled data for the outer objective. Meta-Semi [4] uses the labeled data for the inner objective and unlabeled data for the outer objective. Here the labeled data and unlabeled data refer to the $\mathcal{D}^l$ and $\mathcal{D}^u$ for the SSL. Naturally, these methods deploy the data by using different specific loss functions.
> >
> >
> > Bi-level optimization is an optimization framework, whereas meta-learning is a specific learning setting. While bi-level optimization can often be conveniently employed for meta-learning, it's also applicable in non-meta learning contexts.  Meta-learning is more often associated with scenarios where the learning and validation sets are completely different.

---

> > > ### Comment · Reviewer_akKq · 2023-11-22
> > >
> > > Thanks for addressing my concern. I would keep my score.

---

> > > > ### Author Response · Authors · 2023-11-23
> > > >
> > > > We are glad that we have addressed your concerns and would like to take this opportunity to thank you for your insightful comments and engagement, which have been very helpful in improving our paper.

---

### Official Review · Reviewer_Yzuc · 2023-10-30

**Soundness:** 2 fair
**Presentation:** 3 good
**Contribution:** 2 fair
**Rating:** 5
**Confidence:** 4

**Summary:**

This paper proposes a Bi-level Optimization method for Pseudo-label Learning (BOPL) for Semi-Supervised Learning, which treats pseudo-labels as latent variables and formulates pseudo-labeling as a bi-level optimization problem to jointly learn the pseudo-labels and model parameters within a bi-level optimization framework. This approach simultaneously enhances the quality of pseudo-labels and the prediction consistency between labeled and unlabeled data. Experimental results validate the effectiveness of the proposed approach and show that it outperforms the state-of-the-art SSL techniques.

**Strengths:**

1. This paper defines the SSL problem as a novel bi-level optimization problem, which directly learns the pseudo-labels of unlabeled samples as latent variables through an outer optimization, updates pseudo-labels using teacher-student model, and learns the model parameters through an inner optimization.
2. BOPL employs a pair of bi-level objectives to ensure prediction consistency between labeled and unlabeled samples and combines Interpolation Consistency Training (ICT) to facilitate model training and improve the robustness and generalizability of the model.
3. The experiments are compared with existing SSL methods, and ablation study and sensitivity analysis are conducted. The results demonstrate the effectiveness of BOPL.

**Weaknesses:**

1. The pseudo-labels are simply initialized by the initial teacher model parameters and the update process is impacted by various factors, including the learning rate α, linear combination weight γ and the EMA based teaching models. Consequently, the update of the pseudo-labels tends to be slow. Are there alternative methods that can be used to initialize pseudo-labels to expedite the convergence of model training, such as employing warm-up with labeled data or using mean soft labels (1/num_classes)?
2. In the Algorithm 1, the update of $\theta^{t+1}_{S}$ in the last 3rd line is equivalent to the 7th line and it seems to be redundant.
3. Both Meta Pseudo-Labels[1] and Meta-Semi[2] formulate SSL as a bi-level optimization problem, please explain in detail the differences between BOPL and them.
4. In Table 1, there exists some unreasonable data, which is different from the original paper. For example, in ReMixMatch[3], the test error of 6.27±0.34 for 250 labels and 5.14±0.04 for 4000 labels in CIFAR-10 in original paper differs from the values presented in this paper. Is it because the experimental setup is different from the original paper? Furthermore, in BOPL, the test error for 250 labels is lower than that for 1000 labels in CIFAR-10, which seems to contradict common sense. If these results are accurate, it could imply that BOPL may not be the top-performing method under this setting. According to common expectations, SimMatch's [4] results at 1000 labels should fall within the range of 4.84 and 3.96, while BOPL's reported result of 5.12 seems less favorable by comparison.
5. Compared with existing SSL methods, it's possible that the experiments lack some widely used and important settings, particularly in more challenging conditions. It is suggested to validate the effectiveness of the BOPL approach under the setting of 40 labels in CIFAR-10, 400 labels in CIFAR-100 and 250 labels in SVHN, following the setups of FixMatch[5] and SimMatch[4]. Meanwhile, the results of BOPL+ICT are missing from the main text, please provide additional experimental data.

[1]Hieu Pham, Zihang Dai, Qizhe Xie, and Quoc V Le. Meta pseudo labels. In Proceedings of the IEEE/CVF Conference on Computer Vision and Pattern Recognition (CVPR), 2021.
[2]Yulin Wang, Jiayi Guo, Shiji Song, and Gao Huang. Meta-semi: A meta-learning approach for semi-supervised learning. arXiv preprint arXiv:2007.02394, 2020.
[3] David Berthelot, Nicholas Carlini, Ekin D Cubuk, Alex Kurakin, Kihyuk Sohn, Han Zhang, and Colin Raffel. ReMixMatch: Semi-supervised learning with distribution matching and augmentation anchoring. In International Conference on Learning Representations (ICLR), 2020.
[4] Mingkai Zheng, Shan You, Lang Huang, Fei Wang, Chen Qian, and Chang Xu. Simmatch: Semisupervised learning with similarity matching. In Proceedings of the IEEE/CVF Conference on Computer Vision and Pattern Recognition (CVPR), 2022.
[5] Kihyuk Sohn, David Berthelot, Nicholas Carlini, Zizhao Zhang, Han Zhang, Colin A Raffel, Ekin Dogus Cubuk, Alexey Kurakin, and Chun-Liang Li. Fixmatch: Simplifying semi-supervised learning with consistency and confidence. Advances in neural information processing systems (NeurIPS), 2020.

**Questions:**

Please respond to the questions mentioned above.

---

> ### Author Response · Authors · 2023-11-20
> **Response to Reviewer Yzuc**
>
> We sincerely appreciate the time and effort the reviewer has dedicated to reviewing our work.
> * **About The pseudo-labels initialization**
>
> As the initial pseudo-labels serve as the starting point for the bi-level optimization, it is beneficial to start with good initial pseudo-labels to accelerate the training process. That is the very reason we proposed to pre-train teacher and student models on the training data (labeled and unlabeled) using the Mean-Teacher approach and use them as the initial teacher and student models (please see the description under Eq.(1) in the paper). The pseudo-labels are then initialized by the pre-trained initial teacher model, which are much more informative than using mean soft labels.  One may use other pre-training methods to provide the initial pseudo-labels. But our choice of pre-training with the Mean-Teacher approach can conveniently provide the initial student and teacher model parameters as well.  In the experiments, our proposed method substantially outperforms the Mean Teacher method.
>
> * **About redundant lines in Algorithm 1**
>
> The 7th line in Algorithm 1 updates the model parameters $\underline{\text{on the current minibatch}}$ using $\underline{\text{pseudo-labels in the current step}}$, making the model ready to calculate the outer loss, while the last 3rd line from the end employs $\underline{\text{updated pseudo-labels}}$ for the final update on the model parameters $\underline{\text{on } \mathcal{D}^u}$ at the end of the iteration. As a result, each line serves a distinct function in the model's iterative refinement process.
>
> * **About details of  Meta Pseudo-Labels[1] and Meta-Semi[2] and the differences between BOPL and them**
>
> The three methods formulate SSL as distinct bi-level optimization problems which have different bi-level optimization variables and objectives, hence different bi-level optimization subproblems.
>
> **Meta Pseudo-Labels**: At the **outer level, the teacher network's parameters** are optimized, while at the **inner level**, the **student network's parameters** are adjusted using pseudo-labels generated by the teacher network.
>
> **Meta-Semi**: It adjusts the **weights of unlabeled instances at the outer level**, guided by the loss on labeled samples. At the **inner level**, it learns **model parameters** by minimizing the weighted loss on unlabeled samples using predicted pseudo-labels.
>
> **BOPL**: It uses an **inner loss to optimize model parameters**, utilizing unlabeled data with pseudo-labels. The **outer loss optimizes the pseudo-labels**, treated as latent variables, by using both labeled and unlabeled samples; here, the model parameters are considered a function of the pseudo-labels.
>
> * **About comparison results**
>
> We cited the ReMixMatch’s results reported in the SimMatch paper since our experimental setup is the same as the SimMatch paper.
> BOPL  produced smaller errors with 250 labels than 1000 labels. It could be due to poor local optima. Our full approach, BOPL+ICT overcomes this issue, achieving errors 4.12 (for 250 labels) and 4.02 (for 1000 labels).
> As for comparing SimMatch with 1000, we compared our method with the reported results of SimMatch. SimMatch’s result for 1000 labels on CIFAR-10 was not reported.
>
> * **About adding BOPL+ICT experiments results**
>
> We included additional experimental results as well as the BOPL+ICT results in Table 5 and Table 6 in the appendix. We have added the BOPL+ICT results in the main text of the revised paper.

---

> > ### Comment · Reviewer_Yzuc · 2023-11-21
> >
> > Most of my apprehensions have been resolved by the author. I anticipate their further revision and enhancement of the manuscript based on the recommendations I offered for subsequent versions.

---

> > > ### Author Response · Authors · 2023-11-23
> > >
> > > We are glad that we have addressed your concerns and are grateful for the insightful comments. Your feedback has been very helpful in enhancing the quality of our paper.

---

### Official Review · Reviewer_hAat · 2023-10-30

**Soundness:** 3 good
**Presentation:** 2 fair
**Contribution:** 2 fair
**Rating:** 5
**Confidence:** 3

**Summary:**

This paper proposes a new method to update pseudo-labels commonly used in semi-supervised learning (SSL). The "bi"-level optimization refers to one objective being the training loss with pseudo-labels as targets, which is a function of both parameters and pseudo-labels, and the other objective of refining pseudo-labels. The objective for pseudo-labels is defined to be the cross entropy on the labeled dataset + entropy of model predictions on the unlabeled dataset. The authors provide a cheap approximation to the gradients of this objective, which is optimized with projected gradient descent, where the authors experiment with different projections (ReLU vs. softmax). This procedure is combined with other SSL methods and tested on common SSL datasets.

**Strengths:**

* The proposed method outperforms many other SSL algorithms on common SSL benchmark datasets.
* Ablation studies shows that pseudo-labels benefit from ReLU projections instead of softmax which is an obvious choice for projection onto the probability simplex. It's good that the paper experiments with projection methods and found that there is a better working solution that softmax.

**Weaknesses:**

* The approximation in Eq. 14 still requires two parameter updates to compute the gradient; this must be slow.
* Presentation could be improved. Prop. 1 is just a nice expression for the chain rule - consider combining with proposition 2 or moving it to the Appendix? I also think the description in Alg. 1 could be simplified by writing only the steps critical to the algorithm, e.g. no need to write "compute loss L_{inner}" which doesn't really help with understanding. Or if the authors feel that it's needed, write the expression again in the algorithm description to make it self-contained.
* The novelty of this method seems incremental - while it's practically useful, the only real contribution is to write the parameters $\theta_S$ as a function of $Y$ with a cross-entropy / entropy loss, from which the method naturally follows. However it's not clear why this method results in improved performance.

**Questions:**

Is the pseudo-label updates implemented using expression in Eq. (14) under Prop. 2 to update the labels? It isn't mentioned for sure whether this is used; Equation 17 suggests that the labels are being updated with the true gradients rather than the approximation form in Eq. 14. If so, why do the authors describe the approximation?

---

> ### Author Response · Authors · 2023-11-20
> **Response to Reviewer hAat**
>
> We sincerely appreciate the time and effort the reviewer has dedicated to reviewing our work.
>
> * **About the approximation in Eq. 14**
>
>  Firstly, in Eq.(14), the computation of the parameters $\theta_S^+$ and $\theta_S^-$ actually only requires one **single** standard gradient calculation on the outer loss $\mathcal{L}\_\text{outer} $  for $\delta{\theta\_S^{t+1}}$. The extra cost for computing $\nabla\_{\widehat{\bf y}\_i} \mathcal{L}\_\text{outer}$  is only on computing two predictions on one  unlabeled instance ${\bf x}^u\_i$ using ${\bf \theta}\_S^+$ and ${\bf \theta}\_S^-$ respectively. Therefore, the extra computational cost over standard gradient descent is not significant. Secondly, direct computation of the target gradient $\nabla\_{\widehat{\bf y}\_i} \mathcal{L}\_\text{outer}$  is very difficult. The approximation in Eq.(14) provides a convenient solution in this scenario.
>
> * **About Presentation**
>
> Thanks for the suggestions. We use two separate propositions to enhance clarity in illustrating the rationale behind the two separate key steps. Aligning the loss terms in the algorithm with their corresponding equations is also intended to ensure clarity and avoid confusion.
>
>
> * **About novelty**
>
> The proposed BOPL approach provides a novel perspective and an effective solution to semi-supervised learning by treating pseudo-labels as latent variables and formulating pseudo-labeling as a bi-level optimization problem. Bi-level optimization offers a systematic optimization framework for coherently learning two sets of inter-dependent variables through a pair of bi-level objectives.  Our bi-level optimization formulation allows us to optimize the pseudo-labels interdependently with the model parameters in a coherent and simultaneous manner, avoiding problems, such as cumulative errors and inconsistency, often encountering in separate-step optimization methods.  The effectiveness of our approach is demonstrated through extensive experiments on various benchmarks, showcasing its practical utility.
>
>
> * **About pseudo-label updates**
>
>  The pseudo-label updates are indeed implemented using the expression in Eq. (14) under Proposition 2 to update the labels. It is very difficult to directly compute the (true) gradient, which is the reason we proposed the approximation in Eq.(14).  The gradient term used in Eq. (17) is the term in Eq.(14). We added a sentence in the paper to confirm this.

---

> > ### Comment · Reviewer_hAat · 2023-11-22
> >
> > * Approximation in Eq. 14: Yes, the parameters $\theta^+, \theta^-$ are updated using a single $\delta$ but as written, the gradient of loss then requires a forward pass on the same sample twice with these different parameters. Having to compute $\delta$ and two different forward passes for each sample would increase training time or memory by at least a factor of 2.
> > * Novelty: The authors argue that the proposed method does not suffer from cumulative errors and inconsistency. The proposed method achieves small gains with respect to existing methods, but I don't see how it's clear that the proposed method this is a result of resolving these two issues.

---

> ### Author Response · Authors · 2023-11-23
>
> * **About approximation**
>
> In bi-level optimization problems on deep neural networks, computing second-order derivatives is very complex and that’s why finite difference approximation is common practice [1]. Most pseudo-labeling methods compute their pseudo-labels on each iteration, using one forward pass on unlabeled data. It is important to note, 2 forward passes (that is used in BOPL) do not increase memory requirements compared to 1 forward pass. This is because these passes are executed sequentially and independently, not requiring additional concurrent memory allocation.
>
> Moreover, the following table shows GPU (NVIDIA GeForce RTX 3060) and CPU times for a single iteration (over all the mini-batches) using a batch size of 32 on the WRN-28-2 backbone for different methods. Although the two forward passes add to the training time, the added time is not significant compared to other methods. $\underline{\text{Given the current availability of computational resources, the computational requirement of our method should not pose any challenge.}} $
>
> | Method      | CPU time | GPU time   |
> |-------------|----------|------------|
> | UDA         | 5.084s   | 164.989ms  |
> | CoMatch     | 5.154s   | 208.069ms  |
> | BOPL        | 6.808s   | 250.572ms  |
>
>
> * **About novelty**
>
> We adopt bi-level optimization to address SSL through a novel and direct pseudo-labeling design.
> Specifically, we treat pseudo-labels as latent variables and model parameters as functions of the pseudo-labels, and formulate pseudo-labeling as a bi-level optimization problem to jointly learn the pseudo-labels and  model parameters at different levels through a pair of bi-level objectives. There are clear advantages of using Bi-level optimization over updating the model and pseudo-labels alternatively. By optimizing the pseudo-labels directly, the bi-level optimization method is expected to produce more reliable labels for the unlabeled samples that directly support the optimal model parameters, since both pseudo-labels and model parameters are learned interdependently.
>
>
> This is further evidenced by its performance. Compared to the state-of-the-art method SimMatch, BOPL achieves a 0.84% improvement on CIFAR-10 with 4,000 labeled instances. While this gain may appear modest, it's noteworthy our method has already achieved **a very low error rate (BOPL: 3.12%), where any additional improvement is inherently limited**. Note, SimMatch only achieves a 0.23% improvement over the previous FlexMatch method on the same dataset setting (CIFAR-10 with 4,000 labeled instances), which is much smaller than the performance gain our method achieved with an even lower error rate. Comparisons in other dataset settings are similar. For example, SimMatch only surpasses CoMatch by 0.07% on CIFAR-10 with 250 labeled data, whereas BOPL’s improvement over SimMatch is 0.19% under the same conditions.
>
> Furthermore, unlike SimMatch, which does not consistently surpass previous methods (e.g., Meta Pseudo-labels on CIFAR-10 with 4,000 labeled instances), BOPL demonstrates consistent superiority over current state-of-the-art techniques. In addition, our proposed full approach BOPL+ICT further boosts the performance. This highlights our proposed method’s substantial performance enhancement for SSL.
>
>
>
>
>
> [1] Hanxiao Liu, Karen Simonyan, and Yiming Yang. DARTS: Differentiable architecture search. In International Conference on Learning Representations (ICLR), 2019

---

> > ### Author Response · Authors · 2023-11-23
> >
> > We hope our response has addressed your questions and would like to take the time to thank the reviewer for providing a detailed review and engaging in discussion.

---

### Meta-Review · Area_Chair_JJfw · 2023-12-05

**Metareview:**

This paper proposes a method to enhance semi-supervised training through a bi-level optimization that optimizes the model parameters and pseudo-labels jointly. The proposed framework is interesting and reasonable. However, there are still some concerns regarding the novelty and significance. The use of bi-level optimization techniques has already been widely employed in rectifying incorrect labels, and the current proposal does not provide many new insights to the SSL community. Moreover, bi-level optimization introduces additional computational overhead, but the performance improvement it brings is limited. Further discussions on computational costs or methods for acceleration can further enhance this work. Therefore, I recommend rejecting this paper, but I encourage the authors to use feedback from reviewers to further improve this work.

**Justification For Why Not Higher Score:**

The reason for not assigning a higher score is due to the following shortcomings identified in this paper:
1. Bi-level optimization requires much more computational resources, but its performance improvement is limited. More efforts on reducing computational overhead and accelerating convergence can further enhance the practicality of this proposal.
2. The use of bi-level optimization techniques to rectify mislabeling has already been widely employed in many label noise learning algorithms. Therefore, the algorithm itself may not be considered highly novel.

**Justification For Why Not Lower Score:**

N/A

---

### Decision · Program_Chairs · 2024-01-16

Reject